# The Expression of Cellular Prion Protein, PrPC, Favors pTau Propagation and Blocks NMDAR Signaling in Primary Cortical Neurons

**DOI:** 10.3390/cells12020283

**Published:** 2023-01-11

**Authors:** Rafael Rivas-Santisteban, Iu Raïch, David Aguinaga, Carlos A. Saura, Rafael Franco, Gemma Navarro

**Affiliations:** 1Centro de Investigación Biomédica en Red Enfermedades Neurodegenerativas (CiberNed), National Institute of Health Carlos III, 28029 Madrid, Spain; 2Departament de Bioquímica i Biomedicina Molecular, Universitat de Barcelona, 08028 Barcelona, Spain; 3Institute of Neuroscience (NeuroUB), University of Barcelona, Av. Joan XXIII 27-31, 08028 Barcelona, Spain; 4Department de Bioquímica i Biologia Molecular, Universitat Autònoma de Barcelona, 08193 Barcelona, Spain; 5Department of Biochemistry and Physiology, Faculty of Pharmacy and Food Science, University of Barcelona, 08028 Barcelona, Spain; 6School of Chemistry, University of Barcelona, 08028 Barcelona, Spain

**Keywords:** Alzheimer’s disease, NMDA, prion protein, Tau protein, pTau protein, axonal transport

## Abstract

Background: The N-methyl-D-aspartate receptor (NMDAR) is a target in current treatments for Alzheimer’s disease (AD). The human prion protein (PrPC) has an important role in the pathophysiology of AD. We hypothesized that PrPC modulates NMDA signaling, thus being a process associated with Alzheimer’s disease. Methods: NMDAR signaling was characterized in the absence or presence of PrPC in cAMP level determination, mitogen-activated protein kinase (MAPK) pathway and label-free assays in homologous and heterologous systems. Bioluminescence resonance energy transfer was used to detect the formation of NMDAR-PrPC complexes. AXIS™ Axon Isolation Devices were used to determine axonal transport of Tau and pTau proteins in cortical primary neurons in the absence or presence of PrPC. Finally, proximity ligation assays were used to quantify NMDA-PrPC complex formation in neuronal primary cultures isolated from APP_Sw/Ind_ transgenic mice, an Alzheimer’s disease model expressing the Indiana and Swedish mutated version of the human amyloid precursor protein (APP). Results: We discovered a direct interaction between the PrPC and the NMDAR and we found a negative modulation of NMDAR-mediated signaling due to the NMDAR-PrPC interaction. In mice primary neurons, we identified NMDA-PrPC complexes where PrPC was capable of blocking NMDAR-mediated effects. In addition, we observed how the presence of PrPC results in increased neurotoxicity and neuronal death. Similarly, in microglial primary cultures, we observed that PrPC caused a blockade of the NMDA receptor link to the MAPK signaling cascade. Interestingly, a significant increase in NMDA-PrPC macromolecular complexes was observed in cortical neurons isolated from the APP_Sw,Ind_ transgenic model of AD. Conclusions: PrPC can interact with the NMDAR, and the interaction results in the alteration of the receptor functionality. NMDAR-PrPC complexes are overexpressed in neurons of APP_Sw/Ind_ mouse brain. In addition, PrPC exacerbates axonal transport of Tau and pTau proteins.

## 1. Introduction 

Alzheimer’s disease (AD) is the most prevalent type of dementia and has age as the main risk factor [1]. Unfortunately, the therapeutic tools available to address symptoms are not very efficacious, whereas there is not any treatment to delay disease progression. Among the few approved medicines, memantine (marketed in the US as Namenda^®^) is a negative allosteric modulator of N-Methyl-D-Aspartate glutamate receptors (NMDARs) [2]. These ionotropic receptors are key for neurotransmission in the central nervous system (CNS) of mammals, but they are also expressed in glial cells [3,4]. The NMDA receptor is blocked by Mg^2+^ that is removed by neuronal depolarization. Thus, the NMDAR could be considered a detector of postsynaptic depolarization by binding glutamate and inducing sodium and calcium release [5]. However, there are both postsynaptic and presynaptic NMDARs.

The two main pathological features of AD are the extracellular deposits of ß-Amyloid and intracellular neurofibrillary tangles, which are aggregates containing aberrantly phosphorylated Tau microtubule-associated protein [6,7]. Quite a number of neurodegenerative diseases are associated with aggregates that directly and/or indirectly affect neuronal survival. Thus, in the brain of Parkinson’s disease patients, there are aggregates containing alpha-synuclein [8], while in the brain of Huntington’s disease patients, there are aggregates of mutant poly-glutamine huntingtin [9]. All three diseases are different pathological entities, but all exhibit dementia that worsens with age and protein aggregation [10]. There are more dementias associated with protein aggregation; these are known as prion diseases or transmissible spongiform encephalopathies, and the most common is Creutzfeldt–Jakob disease [11]. Prions are proteins that in their native conformation perform a physiological function, but upon structural changes aggregate and produce neuronal death [12]. On the one hand, it has been demonstrated that (i) genetic deletion of PrNP (the gene encoding for PrPC) reversed hippocampal synapse loss and completely rescued pre-existing behavioral deficits in APPswe/PS1ΔE9 mice transgenic model of AD [13]; (ii) the binding of a specific antibody, AZ59, to PrPC blocks the binding of ß-amyloid to PrPC protein. Consequently, AZ59 treatment in aged symptomatic APP/PS1 transgenic mice re-establishes behavioral and synaptic loss [14]. On the other hand, it has been hypothesized that AD progression is linked to the misfolding and self-assembling of ß-amyloid and Tau proteins in AD and alpha-synuclein in PD, which behave like prions [15].

The aim of this paper was to investigate the potential effects on the NMDAR function of proteins prone to aggregate in neurodegenerative diseases. The results suggest an interaction between the PrPC prion protein and the NMDAR. A direct interaction was found in a heterologous expression system expressing the NR1 and NR2B subunits of the NMDAR and the prion protein and in neuronal primary cultures transfected with a plasmid containing the coding sequence of the prion PrPC protein. In addition, it was shown that the level of NMDAR-PrPC formation of macromolecular complexes was increased in primary neurons from the APP_Sw,Ind_ transgenic AD mouse model. 

## 2. Materials and Methods 

### 2.1. Reagents

N-methyl-D-aspartate (NMDA) and MK-801 were purchased from Tocris Bioscience (Bristol, UK). alpha-synuclein fibrils were prepared by shaking purified recombinant alpha-synuclein as previously described (Masuda-Suzukake et al., 2014; Tarutani et al., 2016a) and Tau and pTau proteins were kindly provided by Prof. J. Avila (CBM, UAM-CSIC, Madrid, Spain). 

### 2.2. Expression Vectors

cDNA for the human version of the GluN1 subunit of the NMDAR lacking the stop codon was obtained by PCR and subcloned to RLuc-containing vector (pRLuc-N1; PerkinElmer, Wellesley, MA, USA) using sense and antisense primers harboring unique restriction sites for HindIII and BamHI; the generated cDNA encodes a GluN1Rluc fusion protein. cDNA for the human version of the GluN2B subunit of the NMDAR was subcloned in pcDNA3.1. cDNA for the human version of PrNP was subcloned in pcDNA3.1. In HEK-293T functional assays, both cDNAs encoding for GluN1 and GluN2B were cotransfected to constitute a tetrameric functional NMDAR.

### 2.3. Cell Culture

HEK-293T cells were grown in Dulbecco’s modified Eagle’s medium (DMEM) supplemented with 2 mM L-glutamine, 100 U/mL penicillin–streptomycin and 5% (*v*/*v*) heat-inactivated fetal bovine serum (FBS) (Invitrogen, Paisley, Scotland, UK).

To prepare mice cortical and hippocampal primary microglial cultures, brain was removed from C57/BL6 mice between 2 and 4 days old. Microglia cells were isolated as described in [16] and plated at a confluence of 40,000 cells/0.32 cm^2^ and grown in DMEM medium supplemented with 2 mM L-glutamine, 100 U/mL penicillin–streptomycin and 5% (*v*/*v*) heat-inactivated FBS (Invitrogen, Paisley, Scotland, UK) for 12 days.

Neuronal primary cultures were prepared from the cortex and hippocampus of fetuses from C57/BL6 pregnant mice. Neurons were isolated as described in [17] and plated at a confluence of 40,000 cells/0.32 cm^2^. Neurons were grown in neurobasal medium supplemented with 2 mM L-glutamine, 100 U/mL penicillin–streptomycin and 2% (*v*/*v*) B27 supplement (Gibco; Paisley, Scotland, UK) in a 96-well plate for 12 days. Purity of neuronal primary cultures was determined using ALEXA FLUOR^®^ 488-conjugated anti-NeuN antibody (Abcam, EPR12763). The labeling confirmed that the neuron cultures were 99% pure.

Cells were maintained in a humid atmosphere of 5% CO_2_ at 37 °C.

### 2.4. Cell Transfection

HEK-293T cells were transiently transfected with the corresponding cDNA by means of the poly-ethylenimine (PEI; Sigma-Aldrich) method, as previously described [18]. Briefly, the corresponding cDNA diluted in 150 mM NaCl was mixed with PEI (5.5 mM in nitrogen residues) also prepared in 150 mM NaCl for 10 min. The cDNA-PEI complexes were transferred to HEK-293T cells, which were incubated for 4 h in serum-starved medium. Then, the medium was replaced by fresh supplemented culture medium and cells were maintained at 37 °C in a humid atmosphere of 5% CO_2_. Forty-eight hours after transfection, cells were washed, detached and resuspended in the assay buffer.

### 2.5. Preparation of Human Alpha-Synuclein Fibrils

Alpha-synuclein fibrils were prepared by shaking purified recombinant alpha-synuclein as previously described [19,20]. Briefly, purified recombinant alpha-synuclein (5 mg/mL) containing 30 mM Tris–HCl (pH 7.5), 10 mM DTT and 0.1% sodium azide was incubated for 7 days at 37 °C in a horizontal shaker at 200 rpm, and then ultracentrifuged at 113,000× *g* for 20 min at 25 °C. The pellets were washed with saline and ultracentrifuged as before. The resulting pellets were collected as alpha-synuclein fibrils and resuspended in 30 mM Tris–HCl (pH 7.5). The fibrils were fragmented using a cup horn sonicator (Sonifier^®^ SFX, Branson, Barcelona, Spain) at 35% power for 180 s (total of 240 s, in periods of 30 s on, 10 s off) [20]. Before use, aliquots were left at room temperature and placed in PBS 1× (pH 7.2) at a final concentration of 0.1 µg/µL. These preparations were subjected to 60 pulses of sonication (runtime 30 s: 0.5 s on, 0.5 s off in a BBR03031311digital SONIFIER sonicator). Sonicated fibril preparations were diluted in pre-warmed medium and immediately added to cells.

### 2.6. App_Sw,Ind_ Transgenic Mice

APP_Sw,Ind_ transgenic mice (line J9; C57BL/6 background) expressing human APP695 harboring the FAD-linked Swedish (K670N/M671L) and Indiana (V717F) mutations under the PDGFβ promoter were obtained by crossing APP_Sw,Ind_ to non-transgenic (control) mice. Control and APP_Sw,Ind_ embryos (E16.5) were genotyped individually and used for hippocampal and cortical neurons cultures. Animal care and experimental procedures were carried out in accordance with European and Spanish regulations (86/609/CEE; RD1201/2005). Mice were handled, as per law, by personnel with the ad hoc certificate (issued by the *Generalitat de Catalunya*) that allows animal handling for research purposes.

### 2.7. ERK1/2 Phosphorylation Assays

To determine ERK1/2 phosphorylation, HEK-293T cells expressing GluN1 and GluN2B subunits with or without expressing human PrPC were plated in primary cultures of cortex or hippocampus microglia cells or primary cultures of cortex or hippocampus neurons at a density of 40,000 cells/well in transparent Deltalab^®^ 96-well microplates and kept in the incubator between 1 and 7 days. Two hours before the experiment, the medium was substituted with serum-starved DMEM medium. Then, cells were pretreated or not for 2 h with alpha-synuclein, Tau or pTau proteins at 37 °C followed by treatment at 25 °C for 10 min with the vehicle or antagonist (MK-801) in serum-starved DMEM medium and stimulated for an additional 7 min with the agonist NMDA. Cells were then washed twice with cold PBS before addition of lysis buffer (20 min treatment). A total of 10 microliters of each supernatant was placed in white ProxiPlate 384-well microplates, and ERK1/2 phosphorylation was determined using the AlphaScreen^®^ SureFire^®^ kit (PerkinElmer) following the instructions of the supplier and using an EnSpire Multimode Plate Reader (PerkinElmer, Waltham, MA, USA).

### 2.8. Dynamic Mass Redistribution Label-Free Assays

Cell signaling was explored using an EnSpire^®^ Multimode Plate Reader (PerkinElmer) with label-free technology. Cellular cytoskeleton redistribution movement induced upon receptor activation was detected by illuminating the underside of the plate with polychromatic light and measured as changes in the wavelength of the reflected monochromatic light, which is a sensitive function of the index of refraction. The magnitude of this wavelength shift (in picometers) is directly proportional to the amount of dynamic mass redistribution (DMR). To determine the label-free DMR signal, 10,000 HEK-293T cells were cotransfected with cDNAs for GluN1 (1 µg) and GluN2B (0.75 µg) subunits of the NMDAR with or without cDNA for human PrNP (1 µg). Cells were plated on transparent 384-well fibronectin-coated microplates to obtain 70 to 80% confluent monolayers, and kept in the incubator for 24 h. Before the assay, cells were washed twice with assay buffer (Hanks’ balanced salt solution with 20 mM HEPES, pH 7.15, 0.1% dimethyl sulfoxide) and incubated in the reader for 2 h in 30 μL/well of assay buffer at 24 °C. Then, the sensor plate was scanned, and a baseline optical signature was recorded for 10 min before adding 10 μL of MK-801 NMDAR antagonist, alpha-synuclein, Tau or pTau proteins dissolved in assay buffer for 30 min readings, followed by the addition of 10 μL of NMDA-selective agonist also dissolved in assay buffer. The DMR responses induced by the agonist were monitored for a minimum of 3600 s.

### 2.9. Immunocytochemistry 

Hippocampal neurons were pretreated with Tau (1 µM) or pTau (1 µM) for 48 h before starting the immunolabeling and were maintained at 37 °C in a humid atmosphere of 5% CO_2_. Neurons were treated with NMDA (15 µM) for 30 min before starting the procedure or were transfected with the cDNA for the human PrNP (1 µg) 48 h before starting the immunolabeling. Hippocampal neurons were fixed in 4% paraformaldehyde for 15 min and washed twice with PBS containing 20 mM glycine before permeabilization with PBS-glycine-containing 0.2% Triton X-100 (10 min incubation). Hippocampal neurons were treated for 1 h with PBS containing 1% bovine serum albumin and later labeled with anti-Tau rabbit monoclonal antibody (1/100, Abcam, ab32057) or with anti-pTau (S396) rabbit monoclonal antibody (1/100, Abcam, ab109390) for 1 h and were subsequently washed and incubated with a secondary Cy3-conjugated anti-rabbit IgG antibody (1/200; Jackson ImmunoResearch, West Grove, PA, USA; red) (red).

For the immunostaining of microglia cells to detect the phenotype state, a similar fixation procedure was followed as that for neurons. Microglia cells were labeled with a mouse anti-iNOS monoclonal antibody (1/200; Ref. MA5-17139, ThermoFisher) or with mouse anti-Arginase monoclonal antibody (1/100; Ref. 610708, Biosciences) for 1 h. After that, samples were washed and incubated with a ALEXA FLUOR^®^ 561-conjugated goat secondary antibody (red) (1/200 Jackson ImmunoResearch, West Grove, PA, USA; red) for 1 h.

In all cases, nuclei were stained with Hoechst (1/100 from 1 mg/mL stock; Sigma-Aldrich). Samples were washed several times and mounted with Immu-mount^®^ (ThermoFisher). Images were obtained using a Zeiss LSM 880 confocal microscope (Zeiss, Jena, Germany) with a 63X oil objective.

### 2.10. Cell Viability

Cortical neurons, at DIV 12, were transfected using the PEI protocol with cDNA for human PrNP (1 µg) or the vehicle. After 48 h, neurons were pretreated with alpha-synuclein (4 µM), Tau (1 µM) or pTau (1 µM) for 4 h before starting the cell viability assay and were maintained at 37 °C in a humid atmosphere of 5% CO_2_.

The cell viability assay is based on the principle that live cells maintain intact cell membranes that exclude certain dyes, such as trypan blue 0,4% (Biorad, ref. #1450021). For quantification of live cells, cortical neurons were gently detached and mixed with an equal volume of trypan blue (0.4%). Live cells (%) were counted with a TC20™ Automated Cell Counter (Biorad, 1450102).

### 2.11. Microfluidics assays of Tau Trafficking

Microfluidic standard neuronal devices (with 150 µm microgroove barriers located in the area between the channels; AXIS^TM^ AXon Investigation System, EMD Millipore) were handled following the manufacturer’s protocol.

For neuronal primary cultures, the pure cortical neurons from mouse embryos (E19) were isolated. Before neuron cell seeding, each assembled device was coated with poly-D-lysine (0.1 mg/mL, Gibco A3890401). A total of 10 microliters of cell suspension (5–6 million cells/mL) was added to both chambers of each device by passive pumping. After 30 min of incubation at 37 °C to allow cell attachment, 200 µL of neurobasal medium (supplemented with 2 mM L-glutamine, 100 U/mL penicillin–streptomycin and 2% (*v*/*v*) B27 supplement (Gibco)) was added. Neurons were maintained at 37 °C in a humidified 5% CO_2_ atmosphere. The medium was replaced every three days in each device (a 50 µL difference in media volume was maintained to prevent spontaneous diffusion).

On DIV 10, once the axons fully crossed the microgrooves (150 µm distance) into the axonal compartment of a device, tau or pTau was added into compartment A of each device for 24 h. On DIV 11, human prion protein cDNA (PrNP) was transfected using the PEI protocol or the medium was replaced (control devices). On DIV 14, to detect Tau and pTau, an immunostaining technique was applied using immunocytochemistry protocol. Neurons were labeled with a rabbit anti-Tau antibody (1/100, abcam ab32057) or rabbit anti-phospo-Tau (S396) antibody (1/100, abcam ab109390) and subsequently marked with a Cy3 anti-rabbit (1/200, Jackson InmunoResearch (red)) secondary antibody. Following 2 h of incubation, cells were washed and subsequently imaged after 24 h using a confocal microscope with 40X and 25X objectives (Zeiss LSM 880).

### 2.12. Bioluminescence Resonance Energy Transfer Assay

HEK-293T cells were transiently cotransfected with a constant amount of cDNA encoding GluN1-RLuc (0.75 μg) and GluN2B (1 µg) and with increasing amounts of cDNAs corresponding to PrPC-YFP (0.5 to 2.5 μg). To control the cell number, the sample protein concentration was determined using a Bradford assay kit (BioRad, Munich, Germany) using bovine serum albumin (BSA) dilutions as standards. To quantify fluorescent proteins, cells (20 µg of total protein) were distributed in 96-well microplates (black plates with transparent bottoms) and fluorescence was read in a Fluostar Optima Fluorimeter (BMG Labtech, Offenburg, Germany) equipped with a high-energy xenon flash lamp, using a 10 nm bandwidth excitation filter at 485 nm. For bioluminescence resonance energy transfer (BRET) measurements, the equivalent of 20 µg of total protein cell suspension was distributed in 96-well white microplates with white bottoms (Corning 3600; Corning Inc., Corning, NY, USA). BRET was determined 1 min after adding coelenterazine H (Molecular Probes, Eugene, OR, USA), using a Mithras LB 940 reader (DLReady, Berthold Technologies, Bad Wildbad, Germany), which allows the integration of the signals detected in the short-wavelength filter at 485 nm and the long-wavelength filter at 530 nm. To quantify GluN1-RLuc expression, luminescence readings were obtained 10 min after the addition of 5 µM coelenterazine H. MilliBRET units (mBU) are defined as follows:$$\mathrm{mBU}=\left[\;\frac{\lambda_{530}\left(\mathrm{long}-\mathrm{wavelength}\;\mathrm{emission}\right)}{\lambda_{485}\left(\mathrm{short}-\mathrm{wavelength}\;\mathrm{emission}\right)\;}-C_{f}\;\right]\;\times \;1000$$
where C_f_ corresponds to ((long-wavelength emission)/(short-wavelength emission)) for the RLuc construct expressed alone in the same experiment.

### 2.13. Proximity Ligation Assay

Detection in natural sources of complexes formed by the NMDAR and human PrPC was addressed in primary cultures of hippocampal neurons of wild-type and APP_Sw/Ind_ transgenic mice. Cells grown on glass coverslips were fixed in 4% paraformaldehyde for 15 min, washed twice with PBS containing 20 mM glycine to quench the aldehyde groups, permeabilized with the same buffer containing 0.05% Triton X-100 for 15 min and washed with PBS. Samples were incubated for 1 h at 37 °C with the blocking solution in a preheated humidity chamber. Rabbit polyclonal anti-NMDAR antibody (1/100, ab52177, Abcam, Cambridge, UK) was conjugated with the PLA probe PLUS oligonucleotides and rabbit anti-PrPC antibody (1/100, EP1802Y, Abcam, Cambridge, UK) was conjugated with the PLA probe MINUS oligonucleotides, both using the Duolink^®^ In Situ Probemaker kit (PLUS, DUO92009 or MINUS, DUO92010). After blocking, samples were incubated overnight at 4 °C with a mixture of the NMDAR- and PrPC-conjugated antibodies. Negative controls were performed by omitting the NMDA-conjugated antibody. Ligation and amplification were carried out as indicated by the supplier (Merck), and cells were mounted with Immu-mount^®^ (ThermoFisher). To detect red dots corresponding to NMDAR-PrPC complexes, samples were observed using a Zeiss LSM 880 confocal microscope equipped with an apochromatic 63X oil immersion objective, and 405 and 561 nm laser lines. Nuclei were stained with Hoechst (1/100 from 1 mg/mL stock; Merck). For each field of view, a stack of 2 channels (1 per staining) and 3 Z planes with a step size of 1 µm were acquired. Andy’s algorithm, a specific ImageJ (National Institutes of Health, Bethesda, MD, USA) macro for reproducible and high-throughput quantification of the total PLA foci dots and total nuclei, was used for data analysis.

### 2.14. Data Analysis

All data were obtained from at least five independent experiments and are expressed as the mean ± standard error of the mean (SEM). GraphPad Prism 9 software (GraphPad Inc., San Diego, CA, USA) was used for data fitting and statistical analysis. One-way ANOVA tests with post hoc Bonferroni’s correction was used when comparing multiple values. When a pair of values was compared, Student’s t test was used. Significant differences were considered when the *p* value was *p* < 0.05 (*), *p* < 0.01 (**) or *p* < 0.001 (***).

## 3. Results

### 3.1. Prion Protein Blocks NMDA Receptor (NMDAR)-Induced MAPK Signaling in Cortical and Hippocampal Primary Neurons

By chance, we discovered two seemingly unrelated factors affecting the activation of the MAPK signaling pathway mediated by the NMDAR. On the one hand, we noticed that activation of the MAPK signaling pathway in cortical primary neurons elicited by NMDA was fully reverted by preincubation with proteins related to AD and PD, i.e., by Tau, pTau and alpha-synuclein proteins (Figure 1A). On the other hand, we found that the expression of the prion protein (PrPC), expressed upon transfection with a plasmid containing the ad hoc sequence, led, in cortical neurons, to the inability of NMDA to activate the MAPK signaling pathway (Figure 1A). Both cortical and hippocampal neurons are affected in AD, which is characterized by pTau aggregates and PrPC overexpression. Similar experiments performed in hippocampal primary neurons showed that pretreatment with Tau, pTau or alpha-synuclein or that prion protein expression blocked the NMDA-induced increase in extracellular-signal-regulated (ERK1/2) phosphorylation (Figure 1B). NMDA concentration was determined using dose–response curves performed in cortical and hippocampal primary neurons (Appendix A).

### 3.2. Prion Protein Potentiates NMDA-Induced Neuronal Excitotoxicity

The assays of cortical neuronal death confirmed the excitotoxicity produced by 15 µM NMDA (>50% reduction in viability after 4 h of incubation), while no significant cell death was observed when cortical neurons were treated for 4 h with alpha-synuclein (4 µM), Tau (1 µM) or pTau (1 µM) (Figure 2A). In contrast, the expression of PrPC significantly potentiated NMDA-induced excitotoxicity. In addition, PrPC expression significantly increased neuronal death in cells treated with alpha-synuclein, Tau and pTau (Figure 2A).

Next, we were interested in detecting whether PrPC expression alters Tau and pTau expression levels in cortical neurons. ALEXA FLUOR^®^ 488-conjugated anti-NeuN antibody was used to label neurons (Appendix A). Immunocytochemistry assays in cortical primary neurons expressing the PrPC (Figure 2D,G), revealed a marked decrease in endogenous Tau and pTau expression in comparison with untransfected cells (Figure 2B,E). Finally, we determined whether Tau and pTau expression levels were affected by NMDA (15 µM) treatment. As shown in Figure 2C,F, it was observed that while NMDA treatment decreased Tau expression, it increased pTau levels in cortical primary neurons. Altogether, these results suggest that NMDA treatment favors Tau phosphorylation.

### 3.3. PrPC Expression Potentiates Tau and pTau Propagation in Mice Cortical Neurons

We focused our efforts on analyzing Tau and pTau propagation, with the aim of identifying the mechanisms involved in the neurodegenerative progression that accompanies AD. To do so, we tested the axonal transport of proteins that are relevant in AD physiopathology. Successful experiments were performed using Tau and pTau. We used cortical neurons because the hippocampus was reserved for other experiments due to the approved experimentation protocol that takes into account the 3Rs rule (Replacement, Reduction and Refinement) in the experimentation with animals. Using microfluidic devices (a schematic representation is shown in Figure 3H), we were able to discover that the transport of both Tau and pTau is markedly increased in cortical neurons that are transfected with the plasmid that encodes for the PrPC (Figure 3). Comparable results were obtained when a similar experiment was performed in cortical neurons pretreated with NMDA (15 µM); again, the presence of prion protein increased Tau and pTau propagation (Figure 3G). Thus, it can be concluded that NMDA does not affect prion induction of pTau propagation.

### 3.4. Prion Protein Blocks NMDA Receptor-Induced Signaling in Mice Microglial Cultures

After characterizing the prion PrPC downregulation of NMDAR function in primary cultures of neurons, we moved to perform assays in primary microglia because NMDAR expression in microglial cultures has been reported by different laboratories [3,4]. To look for relationships between the prion protein and the NMDAR and due to the link to the MAPK signaling pathway [21], experiments on ERK1/2 phosphorylation were performed in cortical and hippocampal primary microglia. The results obtained are similar to those obtained in primary neurons. Namely, the NMDA-induced ERK1/2 phosphorylation disappeared in cells preincubated for 2 h with alpha-synuclein (4 µM), Tau (1 µM) or pTau (1 µM) (Figure 4A,B). Additionally, the expression of the PrPC led to results similar to those shown in Figure 1; i.e., it blunted the action of the NMDAR towards activation of the MAPK signaling pathway.

### 3.5. Prion Protein Polarizes Microglia to an M1 Phenotype

Nowadays, it is well established that microglia can show different phenotypes in pathological conditions [22]; using the M0 for resting, M1 (proinflammatory) and M2 (neuroprotective) nomenclature, it is challenging to convert the proinflammatory M1 phenotype into the neuroprotective M2 phenotype. To analyze the prion effect on the microglial phenotype, microglia cells were transfected with the plasmid that encodes for the PrPC and analyzed by immunocytochemistry. In detecting arginase (a marker of the M2 phenotype), it was observed that PrPC induces a drastic decrease in the neuroprotective phenotype (Figure 4C–F). On the other hand, in analyzing iNOS (a marker of the M1 phenotype), the opposite was observed; that is, the proinflammatory M1 phenotype was strongly potentiated by PrPC expression (Figure 4G–J). Finally, the effect of NMDA (15 µM) on microglial polarization was evaluated. Interestingly, NMDA treatment increased both M1 and M2 markers (Figure 4D,H).

### 3.6. NMDA-Induced Function Is Regulated by Prion Protein in Transfected HEK-293T Cells

We went on to investigate the effect of the expression of the PrPC at the qualitative level in a heterologous system, namely HEK-293T cells. These cells do not express functional NMDARs nor PrPC, and thus are an interesting system to analyze the direct role of PrPC in NMDAR signaling without interferences due to other neuronal proteins or receptors. First, the capability of NMDA receptors to transport calcium ions was investigated. in HEK-293T cells transfected with cDNAs for a calcium sensor (see Methods), for NR1 and NR2B and stimulated with NMDA. As expected, a characteristic signal of calcium mobilization was obtained that was completely blocked by pretreatment with the NMDAR antagonist, MK-801 (Figure 5A). Interestingly, incubation for 2 h with Tau or pTau also blocked the NMDA-induced signal. Finally, when HEK-293T cells were transfected with the plasmids coding for the two subunits of the NMDAR and the prion protein PrPC, it was observed that PrPC completely blocked NMDAR signaling (Figure 5B).

In terms of MAPK cascade activation, NMDA-induced ERK1/2 phosphorylation was completely inhibited by pretreatment for 2 h with alpha-synuclein, pTau and the selective NMDAR antagonist, MK-801, and partially blocked by pretreatment with Tau. As expected, alpha-synuclein, Tau, pTau and MK-801 induced no effect on MAPK phosphorylation in the absence of NMDA (Figure 5C). When the same experiment was performed in cells expressing PrPC, the prion protein blocked NMDA-induced signaling (Figure 5D).

Finally, we used a label-free technique, dynamic mass redistribution (DMR), which is mainly used in G-protein-coupled receptor (GPCR) research. Indeed, expression of a functional NMDAR upon transfecting with plasmids encoding for GluN1 and GluN2B subunits led to atypical/negligible DMR signals when NMDA was used both in naïve cells and in cells preincubated with alpha-synuclein, Tau or pTau (Figure 5E). Interestingly, when cells were also expressing the PrPC, NMDA led to time-dependent effects that were significantly enhanced when cells were preincubated with alpha-synuclein or Tau (Figure 5F). All together, these results demonstrate that the prion protein may directly downregulate NMDAR signaling in transfected HEK-293T cells.

### 3.7. NMDAR and PrPC Form Macromolecular Complexes in HEK-293T Cells

To test the possibility that regulation is due to a direct interaction between the NMDAR and the prion protein, bioluminescence resonance energy transfer (BRET) assays were performed in HEK-293T cells expressing a constant amount of NR1-RLuc and NR2B and increasing amounts of PrPC-YFP. Figure 6A shows a saturation BRET curve that indicates that that the NMDAR and PrPC may specifically interact to form macromolecular complexes (BRET_MAX_ 67 ± 6 mBU, BRET_50_ 100 ± 20). A linear nonspecific signal was obtained when the negative control experiment was performed using GHSR1a-YFP instead of PrPC-YFP (Figure 6A).

We complemented those findings with in situ proximity ligation (PLA) assays, which are instrumental for detecting complexes formed by two proteins in natural sources, including brain sections, or in cell cultures [23,24]. Hippocampal primary neurons were treated with selective antibodies against NMDARs and PrPC protein (Figure 6C,D). In Figure 6, panel C shows red dots surrounding the Hoechst-stained nucleus, indicating the existence of NMDAR-PrPC complexes in neurons. When using only the anti-NMDAR antibody, the level of red dots was negligible (Figure 6B).

### 3.8. Interrelationships between NMDAR and the PrPC in a Mice Model of Alzheimer’s Disease (AD)

We used the brain of fetuses from the APP_Sw,Ind_ mouse, which is a well-described model of AD, to isolate hippocampal primary neurons. First, the formation of NMDAR-PrNP complexes was assessed by PLA. As observed in Figure 6C,D, hippocampal neurons from APP_Sw,Ind_ mice showed around 12 red dots/cells, more than doubling the signal observed in cells from control mice. This result indicates that in the AD mice model, there was an important increase in NMDA-PrPC complexes.

Finally, NMDA-induced ERK1/2 phosphorylation was determined in the same primary cultures. In neurons from the brain of both fetuses carrying the Swedish/Indiana APP mutation and those phenotypically characterized as wild-type, NMDA induced an activation of the MAPK signaling pathway that was mediated by the NMDAR because the selective antagonist, MK801, blunted it (Figure 6F,G). Importantly, transfecting neurons with a plasmid encoding for the PrPC led to a lack of effect of NMDA. We then reasoned that these results using cells from the APP_Sw,Ind_ mice could be due to some functional interaction of NMDAR-prion complexes with neuronal proteins (Figure 6F,G, dark blue bars).

## 4. Discussion

In neurons, excess extracellular/interstitial glutamate produces excitotoxicity that is mediated by, among other, NMDARs. This fact is the basis for the proposal and approval of memantine, a negative modulator of the NMDAR, for the treatment of AD. We confirm that PrPC, like other proteins that aggregate in neurodegenerative diseases, is neurotoxic in experiments performed in primary neurons. We observed that the PrPC potentiated NMDA excitotoxicity, something that is a matter of debate, probably due to the experimental setup. Whereas some studies show that prion proteins including mutated versions and fragments increase the excitotoxicity mediated by ionotropic glutamate receptors [25,26], other studies suggest that modulation of NMDA receptors by prion proteins protects against excitotoxicity [27,28,29]. One explanation for these data could be that the differential GluN2 subunits confer on NMDARs distinct ion channel properties and intracellular trafficking pathways [30]. It is well established that GluN2A-containing NMDARs are predominantly expressed in the hippocampus and neocortex while GluN2B-containing NMDARs are more expressed in the striatum. Moreover, GluN2A-containing NMDARs are more frequently located in synaptic sites and GluN2B in extrasynaptic sites [31,32].

Recently, it has been shown that PrPC and copper cooperatively regulate NMDA receptor activity by mediating S-nitrosylation and inhibiting post-translational modification. Consequently, PrPC together with copper protects neurons from excitotoxicity [33]. However, in NMDARs, S-nitrosylation in the hippocampus is greatly reduced when prion strain infection occurs in both the pre-symptomatic and symptomatic stages occurring after prion-induced CNS alterations in mice [28]. Thus, it seems that pathological PrPC strains play an opposite role to that of the physiologic protein in NMDA signaling. In addition, Kudo W. et al. have observed that the PrPC is an essential co-factor in mediating the neurotoxic effect of oligomeric Aβ [34].

One of the main unknowns around AD is the discovery of its cause and the progression of the disease. Different studies posit that the progression of the disease may be due to the spread of Tau or ß-amyloid proteins that act in the same way as prion proteins. First, we observed in cortical neurons that while NMDA treatment decreases Tau expression, it increases pTau levels, indicating that the NMDAR favors Tau phosphorylation. To clarify the role of prions in protein transport, experiments were performed on primary neurons using microfluidic devices. Of relevance was the finding that PrPC modulates the transport of Tau and pTau in primary neurons. The regulation of the propagation of these proteins is interesting as Tau regulates prion gene transcription [35]. Further research is needed to better understand the links between these two types of proteins involved in the pathophysiology of neurodegenerative diseases.

Another relevant finding was the blockade by the PrPC of the link between NMDAR activation and MAPK signaling pathway activation in the hippocampus and also the cortex. In this sense, it has been proven that the mutation of cooper binding sites on PrPC protein counteracts PrPC inhibition of NMDA signaling [36]. In a previous report, we found disruption of the link by alpha-synuclein, Tau and pTau and the involvement of calcium-binding proteins [21]. Specifically, the results show that NMDAR signaling depends on forming complexes with calcium sensors, especially those involving NCS1 that are altered in the APP_Sw,Ind_ mice model of AD.

Microglia are considered the main neuroimmune cells, with three important functions: (i) sensing environmental changes, (ii) promoting neuronal function and well-being and (iii) inducing neuroprotection via neuronal responses [37]. Correlating microglia with prion diseases is highly controversial. On the one hand, prion diseases potentiate microglia activation and also proliferation while inducing astrogliosis and neuronal loss. It has been postulated that PrPC could have an important role in cytokine secretion after microglia activation [38]. In this sense, we have observed that in primary microglia, not only Tau and pTau but also that PrPC expression blunted NMDA-induced signaling.

Nowadays, it is largely accepted that microglia play an important role in all neurodegenerative processes. When it polarizes to an M1 proinflammatory phenotype, it favors pathology progression, while when it polarizes to an anti-inflammatory M2 phenotype, it could prevent or slow down pathology evolution. Our data show that PrPC expression strongly polarized microglia to a proinflammatory M1 phenotype while it induced a drastic decrease in the M2 neuroprotective phenotype.

The NMDAR is prone to multiple regulations, some of them being mediated by interactions with other receptors, e.g., cannabinoid receptors [39] or the adenosine A_2A_ receptor [40]. Different data show that Aß induces neurotoxicity by an abnormal interaction with the prion protein and the glutamatergic receptors mGlu5 and NMDA [41,42]. Along the same lines, it has been described that a trimeric complex formed by PrPC-NMDAR-mGlu5 could regulate Aß’s effects [43]. 

Hence, we reasoned that the effect of the PrPC on NMDAR function regulation could be due to a direct interaction between both proteins, which was proved in a heterologous expression system (Figure 6A) in which the blockade of the NMDAR-MAPK link by PrPC expression was also detected. The PLA was, once more, instrumental in detecting the interaction between the NMDAR and the PrPC, and the findings in primary cultures from the APP_Sw,Ind_ AD model could guide future therapies to combat AD. The increase in the expression of NMDAR/PrPC macromolecular complexes may underlie the events that lead, over time, to cognitive deficits in these transgenic animals. It is tempting to speculate that decreasing the blockade of NMDAR-mediated signaling to the MAPK signaling pathway could be used as a convenient screening assay in order to select more efficacious anti-AD drugs.

## 5. Conclusions

It has been demonstrated that PrPC can complex with the NMDAR, and this results in the blockade of the receptor functionality. In addition, PrPC up-regulates the axonal transport of Tau and pTau proteins and in doing so favors the progression of the pathology. Moreover, NMDAR-PrPC has become a new target in combating AD due to the fact that heteromeric complexes are overexpressed in neurons of APP_Sw/Ind_ animal models.

## Figures and Tables

**Figure 1 cells-12-00283-f001:**
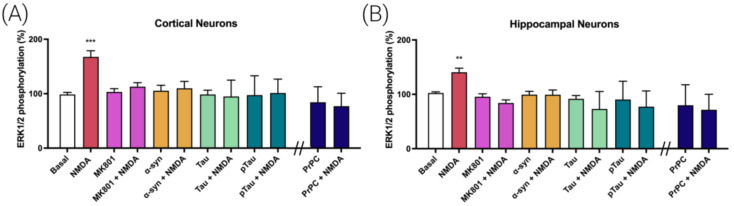
Alpha-synuclein, Tau, pTau and PrPC effect on MAPK signaling pathway activation triggered by NMDA in primary neurons from mice brain. ERK1/2 phosphorylation was analyzed using an AlphaScreen^®^SureFire^®^ kit (Perkin Elmer) in cortical (**A**) or hippocampal neurons (**B**). Primary neurons were treated with vehicle (white) or NMDA (15 µM, red) or preteated with the selective NMDAR antagonist MK-801 (1 µM, purple), alpha-synuclein (4 µM, orange), Tau (1 µM, green) or pTau (1 µM, dark green) prior to NMDA addition (15 µM). Cells transfected with the PrNP (blue, 1 µg) were treated with the NMDA (15 µM) or vehicle. Values are the mean ± S.E.M. of 5 independent experiments performed in triplicate. One-way ANOVA followed by Bonferroni’s multiple comparison post hoc test was used for statistical analysis (** *p* < 0.01, *** *p* < 0.001 versus basal condition). ANOVA summary: (**A**) F: 7.19 *p* < 0.001 (**B**) F: 6.68 *p* < 0.001.

**Figure 2 cells-12-00283-f002:**
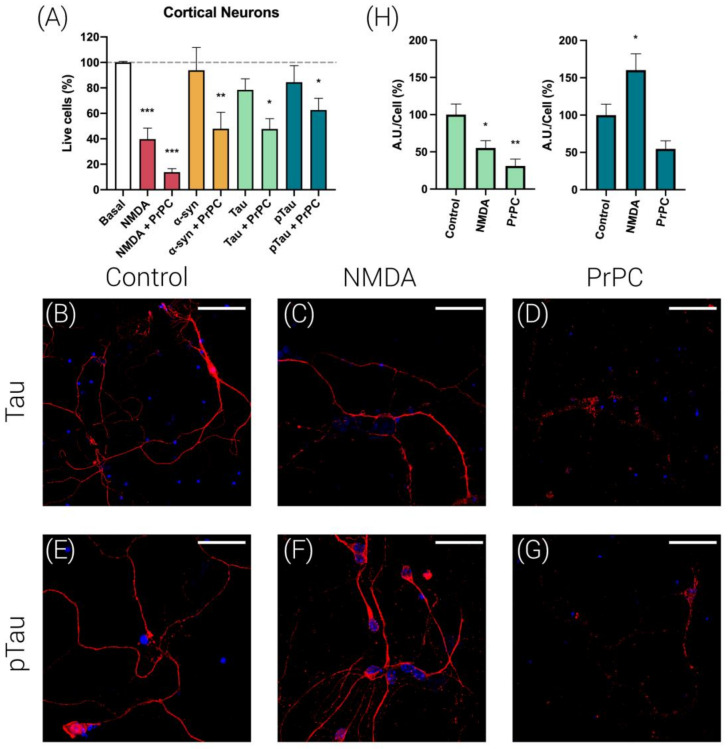
Cell viability assays and Tau and pTau immunolabeling in primary neurons. (**A**) Cortical neurons were transfected or not with the cDNA for the human PrPC protein (1 µg) and subsequently treated with NMDA (15 µM, red), alpha-synuclein (4 µM, orange), Tau (1 µM, green) or pTau (1 µM, dark green) for 4 h. For quantification of living cells, cortical neurons were mixed with an equal volume of trypan blue (0.4%). Bar graph shows live cells (colorless cells/total cells × 100, %) in comparison with basal (untreated cells). Values are the mean ± S.E.M. of 5 independent experiments performed in triplicate. One-way ANOVA followed by Bonferroni’s multiple comparison post hoc test was used for statistical analysis (* *p* < 0.05, ** *p* < 0.01, *** *p* < 0.001 versus basal condition). ANOVA summary: (**A**) F: 8.70 *p* < 0.001. Immunocytochemistry assays were performed in hippocampal primary neurons to detect Tau (**B**–**D**) and pTau proteins (**E**–**G**). Bar graph shows quantification of the amount of fluorescence (Arbitrary Units, A.U.)/cell (%) in comparison with control cells for Tau (green bars) or pTau (dark green bars) (**H**). Values are the mean ± S.E.M. of 5 independent experiments performed in triplicate. One-way ANOVA followed by Bonferroni’s multiple comparison post hoc test was used for statistical analysis (* *p* < 0.05, ** *p* < 0.01 versus basal condition). ANOVA summary: (**H**) (left) F: 8.41 *p* < 0.003. (right) F: 7.88 *p* < 0.003. Tau protein was detected by an anti-Tau rabbit monoclonal antibody and a secondary Cy3-conjugated anti-rabbit IgG antibody (red). pTau protein was detected by an anti-pTau (S396) rabbit monoclonal antibody and a secondary Cy3-conjugated anti-rabbit IgG antibody (red). Neurons were treated with NMDA (15 µM) (**C**,**F**) or were transfected with the cDNA for the human PrNP (1 µg) (**D**,**G**). Cell nuclei were stained with Hoechst (blue). Scale bar: 20 µm.

**Figure 3 cells-12-00283-f003:**
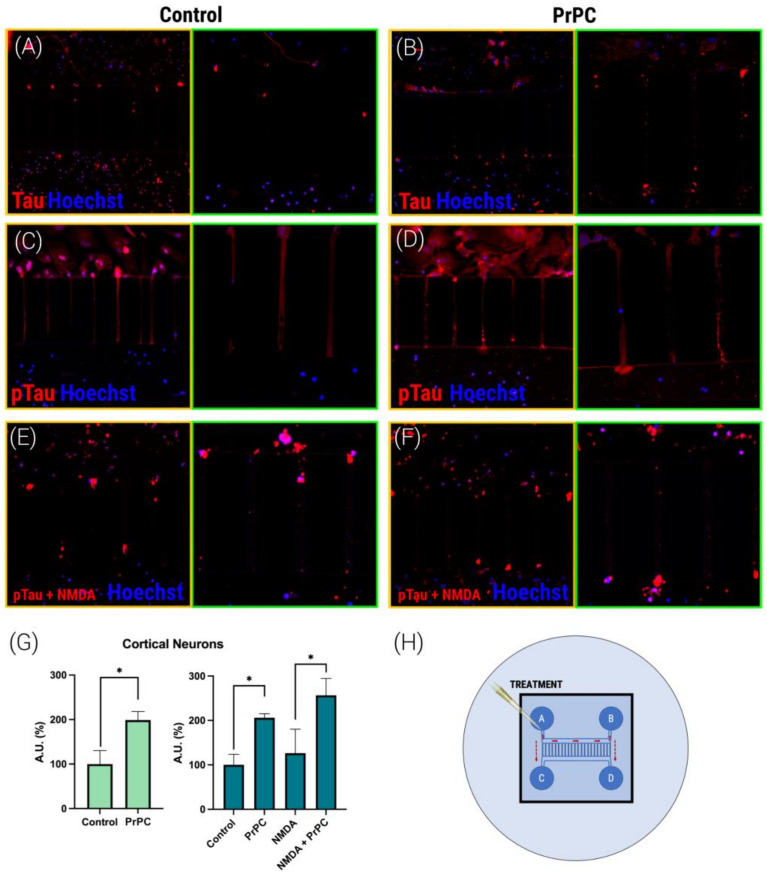
Detection of Tau and pTau axonal transport in mice cortical primary neurons. Mice cortical neurons were grown in the microfluidic device (**A**–**F**). On DIV 10 and once the axons fully crossed the microgrooves (150 µm distance), Tau (1 µM) or pTau (1 µM) proteins were added into compartment A of each device for 24 h. On DIV 11, neurons were transfected with human PrNP cDNA (1 µg). Cells were treated with NMDA (15 µM) (**E**,**F**) or vehicle (basal) (**A**–**D**) for 1 h more. Neurons were labeled with a rabbit anti-Tau antibody (1/100, Abcam ab32057) or rabbit anti-phospo-Tau (S396) antibody (1/100, Abcam ab109390) and subsequently labeled with anti-Cy3 fluorescent rabbit protein (1/200, Jackson InmunoResearch) secondary antibody (red). Following 2 h of incubation, cells were washed and subsequently imaged using a confocal microscope with 25X (yellow squares) and 40X (green squares) objectives (Zeiss LSM 880). (**G**) Bar graph shows quantification of the amount of fluorescence in the microfluidic channel opposite to the channel where proteins were added (Arbitrary Units, A.U.) (%) in comparison with fluorescence in control cells for Tau (green) or pTau (dark green). Values are the mean ± S.E.M. of 5 independent experiments performed in triplicate. One-way ANOVA followed by Bonferroni’s multiple comparison post hoc test was used for statistical analysis (* *p* < 0.05 versus control condition). ANOVA summary: (**G**) F: 4.52 *p* < 0.009. (**H**) A schematic drawing of the microfluidic device and the location of the treatment in the A and B wells, allowing migration through the axons in the microchannels.

**Figure 4 cells-12-00283-f004:**
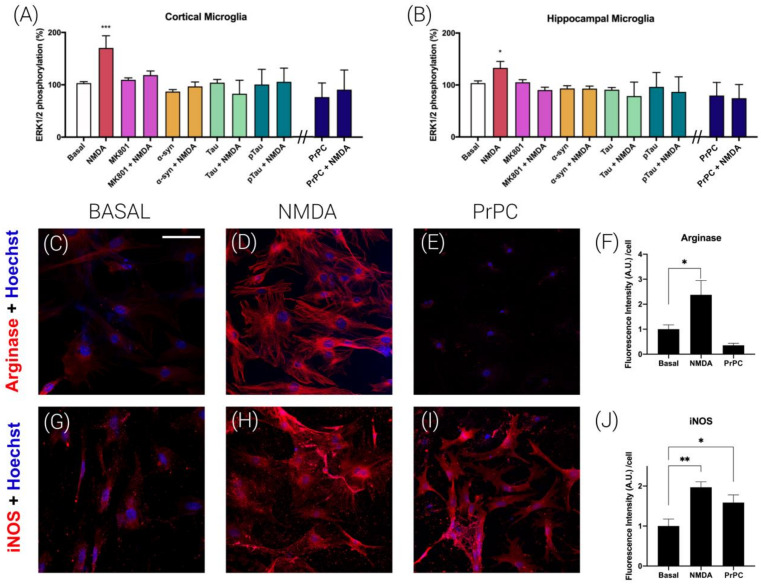
NMDAR-PrPC complexes effect on MAPK signaling in microglial primary cultures from mice. Analysis of proinflammatory (M1) and anti-inflammatory (M2) states of microglia. ERK1/2 phosphorylation was analyzed using an AlphaScreen^®^SureFire^®^ kit (Perkin Elmer) in cortical (**A**) or hippocampal (**B**) microglial cells. Primary cultures were treated with vehicle (white) or NMDA (red), or pretreated with the selective NMDAR antagonist MK-801 (1 µM, purple), alpha-synuclein (4 µM, orange), Tau (1 µM, green) or pTau (1 µM, dark green) prior to NMDA addition (15 µM). Cells transfected with the cDNA for the human PrPC (blue, 1 µg) were treated with NMDA (15 µM) or vehicle. Values are the mean ± S.E.M. of 5 independent experiments performed in triplicate. One-way ANOVA followed by Bonferroni’s multiple comparison post hoc test was used for statistical analysis (* *p* < 0.05, ** *p* < 0.01, *** *p* < 0.001 versus basal condition). ANOVA summary: (**A**) F: 8.36 *p* < 0.001. (**B**) F: 6.38 *p* < 0.001. Immunocytochemistry assays were performed in hippocampal primary microglia. Cells were treated with NMDA (15 µM) or vehicle (basal) for 1 h more. Cells were transfected with human PrPC cDNA (**E**,**I**) (1 µg) or vehicle (basal). The microglia phenotype was characterized by incubating with specific antibodies: anti-Arginase, microglia M2 state (**C**–**F**) or anti-iNOS, microglia M1 state (**G**–**J**). Detection is shown in red due to incubation with a secondary antibody conjugated to ALEXA FLUOR^®^ 561. Bar graphs (**F**,**J**) represent the quantification of the red labeling (fluorescence intensity A.U./cell) in the different groups calculated using Fiji software. Values are the mean ± S.E.M. of 5 independent experiments performed in triplicate. One-way ANOVA followed by Bonferroni’s multiple comparison post hoc test was used for statistical analysis (* *p* < 0.05, ** *p* < 0.01; versus basal condition fluorescence intensity). ANOVA summary: (**F**) F: 12.52 *p* < 0.001. (**J**) F: 25.53 *p* < 0.001. Scale bar: 75 µm.

**Figure 5 cells-12-00283-f005:**
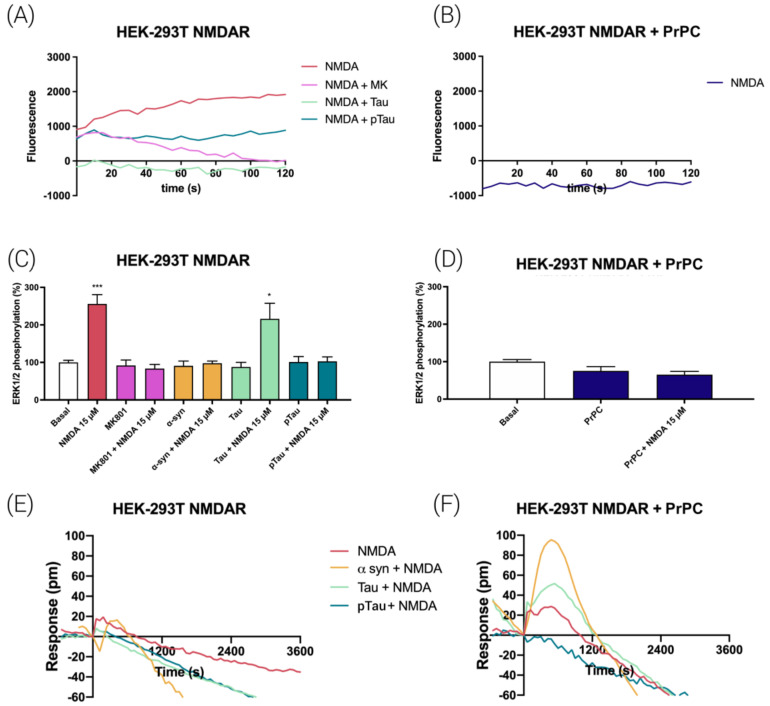
Signaling in HEK-293T cells expressing NMDAR-PrPC complexes. HEK-293T cells were transfected with the cDNAs for two protomers of the NMDA receptor: GluN1 (1 µg) and GluN2B (0.75 µg) and/or with the cDNA for the human PrNP (1 µg) (**B**,**D**,**F**). Cells were treated NMDA (15 µM) and, when indicated, with the selective receptor antagonist, MK-801 (1 µM). Treatments with alpha-synuclein (4 µM), Tau (1 µM) or pTau (1 µM) proteins were performed for 2 h prior to NMDA addition. For calcium mobilization assays, HEK-293T cells were also transfected with the cDNA for the engineered calcium sensor, 6GCaMP (1 µg). Real-time calcium-induced fluorescence was collected for 120 s (**A**,**B**). ERK1/2 phosphorylation was analyzed using an AlphaScreen^®^SureFire^®^ kit (Perkin Elmer) (**C**,**D**). Values are the mean ± S.E.M. of 5 independent experiments performed in triplicate. One-way ANOVA followed by Bonferroni’s multiple comparison post hoc test was used for statistical analysis (* *p* < 0.05, *** *p* < 0.001 versus basal condition). ANOVA summary: (**C**) F: 13.11 *p* < 0.001. (**D**) F: 3.12 *p* < 0.053. DMR readings were collected for 3600 s in cells expressing the NMDAR and in cells expressing the NMDAR and the PrPC prion protein. (**E**,**F**).

**Figure 6 cells-12-00283-f006:**
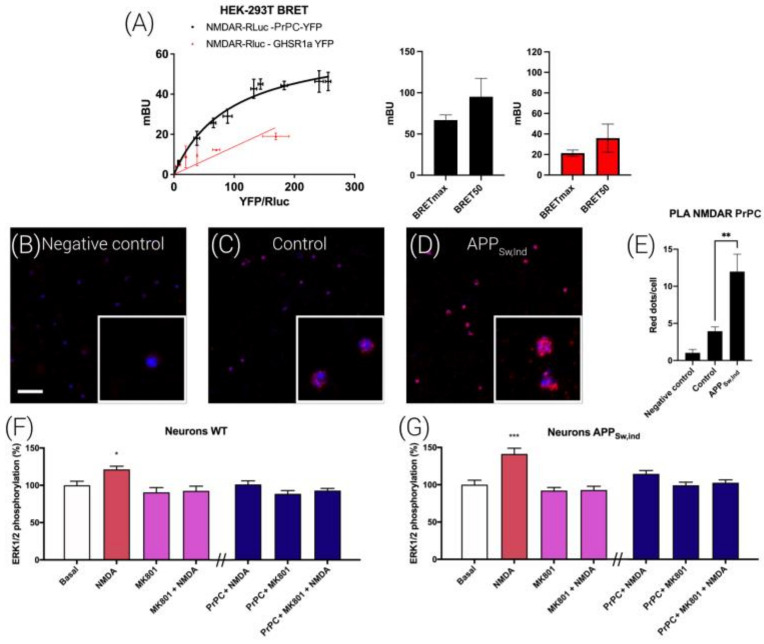
NMDA-PrPC complex expression and functionality in APP_Sw/Ind_ mice hippocampal neurons. (**A**) BRET assays were performed in HEK-293T cells transfected with a constant amount of cDNAs for GluN1-Rluc (0.25 µg), GluN2B (0.15 µg) and increasing amounts of cDNA for PrNP-YFP (0.25 to 1.25 µg) (black) or (as a negative control) GHSR1aR-YFP (0.25 to 1.25 µg) (red). Values are the mean ± S.E.M. of 8 independent experiments performed in duplicate. Expression of NMDA-PrNP complexes in primary neurons (**B**–**D**) of wild-type (**C**) and APP_Sw/Ind_ transgenic mice (**D**) as determined by PLA using specific primary antibodies against the GluN1 subunit linked with the minus probe and against the human PrPC linked with the plus probe. Confocal microscopy images (stacks of 3 consecutive panels) show complexes as red clusters over Hoechst-stained nuclei (blue). As a negative control (**B**), neurons were incubated with the primary antibody against the GluN1 subunit linked with the minus probe only. Cell nuclei were stained with Hoechst (blue). Scale bar: 10 µm. (**E**) Bar graphs show the number of red dots/cells in APP_Sw/Ind_ mice and control animals (** *p* < 0.01; Student’s t test versus the control condition). ERK1/2 phosphorylation was analyzed using an AlphaScreen^®^SureFire^®^ kit (Perkin Elmer) in hippocampal neurons from APP_Sw/Ind_ mice (**G**) and control animals (**F**). Values are the mean ± S.E.M. of 5 independent experiments performed in triplicate. One-way ANOVA followed by Bonferroni’s multiple comparison post hoc test was used for statistical analysis (* *p* < 0.05, *** *p* < 0.001 versus basal condition). ANOVA summary: (**F**) F: 4.18 *p* < 0.001. (**G**) F: 10.14 *p* < 0.001.

## Data Availability

The datasets used and/or analyzed during the current study are available from the corresponding author on reasonable request.

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
