# Peer review of "The Expression of Cellular Prion Protein, PrPC, Favors pTau Propagation and Blocks NMDAR Signaling in Primary Cortical Neurons"

_cells, 2023, doi:10.3390/cells12020283_

Round 1
Reviewer 1 Report
Since NMDA receptor is a therapeutic target for Alzheimer’s disease and human prion protein has a role in this pathology, it was hypothetized that prion protein modulates NMDA receptor signalling in this disorder. The interaction between PrNP and NMDA receptor was characterized in neuronal and glial cell cultures. It was shown that 1. PrNP can complex with NMDA receptor, the complex has reduced receptor activity; 2. PrPN upregulates axonal flow of Tau- and p-Tau proteins, which mirrors progression of this disorder; 3. The complex was overexpressed in Alzheimer’s disease animal model and thus it can form a therapeutic target for this pathology.
Please add prion protein in the title instead of using abbreviation
Fig. 1. Legend, editing should be corrected. Add F value whenever one-way ANOVA was used. Was NMDA effect concentration-dependent on MAPK signalling in cortical and hippocampal neurons?
Fig. 2. Legend needs to be reorganized, indicate text for figure B after figure A. Add statistics used, F value, significance, and n. Also indicate drug concentration for figures A and B, concentration of PrNP used is not shown in the text or legend.
Fig. 3G. These are also missing here, they need to be added in the legend.
Fig 4. Add statistics for bar graphs, F and J figures. It is also incomplete in Fig. 5 C and D.
Author Response
The graphical abstract has been included.
Reviewer 1:
Since NMDA receptor is a therapeutic target for Alzheimer’s disease and human prion protein has a role in this pathology, it was hypothetized that prion protein modulates NMDA receptor signalling in this disorder. The interaction between PrNP and NMDA receptor was characterized in neuronal and glial cell cultures. It was shown that 1. PrNP can complex with NMDA receptor, the complex has reduced receptor activity; 2. PrPN upregulates axonal flow of Tau- and p-Tau proteins, which mirrors progression of this disorder; 3. The complex was overexpressed in Alzheimer’s disease animal model and thus it can form a therapeutic target for this pathology.
Please add prion protein in the title instead of using abbreviation
Fig. 1. Legend, editing should be corrected. Add F value whenever one-way ANOVA was used. Was NMDA effect concentration-dependent on MAPK signalling in cortical and hippocampal neurons?
Thank you for the comment. The Figure legend has been corrected and F value included.
Thank you for the comment. MAPK dose dependent curves in cortical and also hippocampal primary cultures of neurons after NMDA stimulation have been included.
Fig. 2. Legend needs to be reorganized, indicate text for figure B after figure A. Add statistics used, F value, significance, and n. Also indicate drug concentration for figures A and B, concentration of PrNP used is not shown in the text or legend.
Thank you for the comment. Figure legend B has been reorganized. The Figure legend has been corrected and F values included. All drugs concentrations have also been indicated toghether with PrNP concentration.
Fig. 3G. These are also missing here, they need to be added in the legend.
Thank you for the comment. The Figure legend has been corrected and F value and n included.
Fig 4. Add statistics for bar graphs, F and J figures. It is also incomplete in Fig. 5 C d D.
Thank you for the comment. The Figure legend has been corrected and F value included.

Reviewer 2 Report
Alzheimer's disease is a neurodegenerative disorder poorly understood and without an effective treatment. In the manuscript by Santisteban et al, the authors, describe how the prion protein PrNP interacts with NMDAR with the consequent block of its function in primary cultures of neurons and glia cells. This interaction was also observed in vitro by using a heterologous expression system. The results described by the authors are very interesting but need further improvements before being published.
1. The effect of NMDAR-PrNP complexes on MAPK signalling should be improved. Experiment of western blot analysis could be crucial to determinate the ratio of phosphorylated vs unphophorylated forms of ERK1/2, JNK and p38.
2. In Figure 2 the panels (C-H) are very dark and shown only few cells per field.
Author Response
Alzheimer's disease is a neurodegenerative disorder poorly understood and without an effective treatment. In the manuscript by Santisteban et al, the authors, describe how the prion protein PrNP interacts with NMDAR with the consequent block of its function in primary cultures of neurons and glia cells. This interaction was also observed in vitro by using a heterologous expression system. The results described by the authors are very interesting but need further improvements before being published.
- The effect of NMDAR-PrNP complexes on MAPK signalling should be improved. Experiment of western blot analysis could be crucial to determinate the ratio of phosphorylated vs unphophorylated forms of ERK1/2, JNK and p38.
Thank you for the comment. MAPK assays have been developed by kit trying to decrease the number of animals sacrificed in each experiment. Primary neurons are always obtained from mice fetus, then the number of hippocampal neurons use to be really low. The main advantage of the AlphaScreen®SureFire® kit (Perkin Elmer) is the number of cells you require to perform the assay, almost 100 times lower than Western blot. Moreover, the AlphaScreen®SureFire® kit (Perkin Elmer) has been tested by our research group in numerous manuscripts and the results are really confident. Moreover, preparing new primary cultures and testing them by WB it would require a couple of extra months, due to the fact that primary cultures of neurons must be prepared (pregnancy lasts 19 days), grown for a period of two weeks and the WB assay lasts almost a couple of weeks more. However, to improve MAPK signalling assays, MAPK dose dependent curves in cortical and also hippocampal primary cultures of neurons after NMDA stimulation have been included.
- In Figure 2 the panels (C-H) are very dark and shown only few cells per field.
Thank you for the comment. Pannels C-H have been modified to improve the images quality.
Round 2
Reviewer 2 Report
1. The authors did not reply to the request of analysing the single components of the MAPK signaling. The authors’ choice to reduce the number of animals scarified is appreciated but the manuscript needs more convincement experiments to enforce the link between the MAPK signaling and NMDA stimulation. The inclusion of MAPK dose dependent curves of primary cultures after NMDA stimulation, is appropriated and it could be also performed to strength the data regarding the influence of the signaling on M1/M2 polarization. As alternative to the western blot the authors can analyses the activation of single components of MAPK signaling (ERK, JNK and p38) by using MAP kinase kit that detect specific kinase activity.
2. The authors did not reply appropriately to the requirement. They increased the signal intensity of the images but not the number of the cells per field. They should include images with lower magnification.
Author Response
- The authors did not reply to the request of analysing the single components of the MAPK signaling. The authors’ choice to reduce the number of animals scarified is appreciated but the manuscript needs more convincement experiments to enforce the link between the MAPK signaling and NMDA stimulation. The inclusion of MAPK dose dependent curves of primary cultures after NMDA stimulation, is appropriated and it could be also performed to strength the data regarding the influence of the signaling on M1/M2 polarization. As alternative to the western blot the authors can analyses the activation of single components of MAPK signaling (ERK, JNK and p38) by using MAP kinase kit that detect specific kinase activity.
Thank you for the comment. The MAPK phosphorylation assays developed by kit and included in the manuscript detect ERK1/2 phosphorylation and thus, they give information on the MAPK signalling pathway. We have never analyzed p38, nor JNK by kit because nowadays there are no reliable kits to do so. Moreover, to prepare primary cultures and then analyze them we would require almost 2 months due to the fact that females should be pregnant and then neurons should grow for almost two weeks before performing the first test. In this sense, it would be really difficult or impossible to include new data in 10 days.
2. The authors did not reply appropriately to the requirement. They increased the signal intensity of the images but not the number of the cells per field. They should include images with lower magnification.
Thank you for the comment. We have decided to show just few cells due to the fact that it is easier to detect they shape and structure. However, an slide has been submited showing more cells included in the same experiment.
